# Structure and Physical Properties of Mg_93−x_Zn_x_Ca_7_ Metallic Glasses

**DOI:** 10.3390/ma16062313

**Published:** 2023-03-14

**Authors:** Štefan Michalik, Zuzana Molčanová, Michaela Šulíková, Katarína Kušnírová, Pál Jóvári, Jacques Darpentigny, Karel Saksl

**Affiliations:** 1Diamond Light Source Ltd., Harwell Science and Innovation Campus, Didcot OX11 0DE, UK; 2Institute of Materials Research of SAS, Slovak Academy of Sciences, Watsonova 47, 040 01 Košice, Slovakia; 3Institute of Physics, Faculty of Science, Pavol Jozef Šafárik University in Košice, Park Angelinum 9, 041 54 Košice, Slovakia; 4Department of Medical and Clinical Biophysics, Faculty of Medicine, Pavol Jozef Šafárik University in Košice, Trieda SNP1, 040 11 Košice, Slovakia; 5Wigner Research Centre for Physics, Institute for Solid State Physics and Optics, P.O. Box 49, 1525 Budapest, Hungary; 6Laboratoire Léon Brillouin, CEA-Saclay, 91191 Gif sur Yvette, France; 7Faculty of Materials, Metallurgy and Recycling, Technical University of Košice, Letná 9, 042 00 Košice, Slovakia

**Keywords:** metallic glasses, high-energy X-ray diffraction, neutron diffraction, PDF analysis

## Abstract

The Mg-Zn-Ca system has previously been proposed as the most suitable biodegradable candidate for biomedical applications. In this work, a series of ribbon specimens was fabricated using a melt-spinning technique to explore the glass-forming ability of the Mg-Zn-Ca system along the concentration line of 7 at.% of calcium. A glassy state is confirmed for Mg_50_Zn_43_Ca_7_, Mg_60_Zn_33_Ca_7_, and Mg_70_Zn_23_Ca_7_. Those samples were characterised by standard methods to determine their mass density, hardness, elastic modulus, and crystallisation temperatures during devitrification. Their amorphous structure is described by means of pair distribution functions obtained by high-energy X-ray and neutron diffraction (HEXRD and ND) measurements performed at large-scale facilities. The contributions of pairs Mg-Mg, Mg-Zn, and Zn-Zn were identified. In addition, a transformation process from an amorphous to crystalline structure is followed in situ by HEXRD for Mg_60_Zn_33_Ca_7_ and Mg_50_Zn_43_Ca_7_. Intermetallic compounds IM1 and IM3 and hcp-Mg phase are proposed to be formed in multiple crystallisation eventss.

## 1. Introduction

Metallic glasses represent a group of disordered alloys that have attracted the attention of researchers because of their extraordinary features and properties with respect to their crystalline counterparts. One of the recent suggestions is the use of metallic glasses as biomaterials for implants [1,2]. Particularly, an Mg-Zn-Ca system was proposed as the most suitable biodegradable candidate [3]. It consists of the elements Mg, Zn, and Ca pre-existing in a human body that has an inherent tolerance to them [4], with their average quantities in an adult being 25 g, 2 g, and 1100 g, respectively. It has been demonstrated that Mg-based MGs have improved corrosion resistance compared with crystalline Mg-based alloys [5,6,7]. Their compression strength is about 900 MPa [8] and elastic modulus is about 40 GPa [6], closer to elastic modulus of human bones than other groups of biocompatible metallic glasses such as Zr- or Ti-based ones [9]. Many resent studies tailor a composition of the Mg-Zn-Ca system in the range of few percent for Ca (between ∼4 and ∼10 at.%) and Mg in the range from ∼60 to ∼75 at.% [10,11,12]. Specimens containing less than 50 at.% Zn show good corrosion resistance and tolerable cytotoxicity for biomedical applications [13]. To better understand those macroscopical properties, a microstructural description is desired. Unlike crystalline compounds, MGs lack a repetitive motive to produce the long-range ordering. Therefore, standard tools of crystallography are not applicable and different approaches shall be selected.

The amorphous structure of glassy alloys can be characterised by means of atomic pair distribution function analysis. Over years, pair distribution function (PDF) analysis has become a crucial tool in unveiling information about a local atomic arrangement in the short and medium range of MGs [14]. Experimental data necessary for PDF analysis are collected by scattering experiments using different types of radiation (X-rays, neutrons, and/or electrons) [15,16]. Data collection must be performed up to high *Q* values of the scattering vector. In the case of MGs, the structure factor oscillations are usually observable up to 15–20 Å^−1^, but oscillations can persist easily up to 25–30 Å^−1^ or even 45–50 Å^−1^ for oxide glasses [17,18]. Generally, the PDF reflects a probability of finding pairs of atoms separated by a given distance. Details about the terminology and definitions of various types of PDFs can be found elsewhere [19,20,21]. The interpretation of PDFs becomes more complicated when the investigated alloy consists of more than one type of atom. Then, a partial pair distribution function can be introduced defining the distribution of only those atom pairs coming from atoms of type *i* around atoms of type *j*. In principle, the aim is to decompose the total PDF of the alloy under study into its all-possible partials to understand completely the local atomic structure. Describing chemical order in a binary A-B system requires the separation of A-A, A-B, and B-B type partial pair correlations. For ternary and quaternary systems, there are six and ten pair partials contributing to the total PDF, respectively. Consequently, it becomes inevitable to apply various structural techniques to reveal as much as possible about the local atomic arrangement.

The aim of this work is to systematically explore the glass forming ability in the Mg-Zn-Ca system along a Ca composition line fixed at 7 at.% and characterise local atomic structure changes in as-prepared glassy alloys due to a gradual substitution of larger Mg atoms by smaller Zn ones. For that reason, a series of Mg_93−x_Zn_x_Ca_7_ ribbon specimens for x = 3, 13, 23, 33, 43, 53, 63, and 73 alloys was prepared using a melt-spinning technique. Both the total X-ray and total neutron scattering measurements were performed on three glassy alloys, Mg_50_Zn_43_Ca_7_, Mg_60_Zn_33_Ca_7_, and Mg_70_Zn_23_Ca_7_, to probe their local atomic arrangement by means of reduced pair distribution functions. To unambiguously identify the contribution of individual atomic pairs, the simultaneous fitting of the first maximum of the reduced X-ray and neutron pair distribution functions was proposed. In addition, the benefits and limits of using high-energy X-rays and neutron total scattering measurements are demonstrated on the Mg-Zn-Ca system. The PDF curves are present in a form of reduced pair distribution functions, generated via the sine Fourier transformation of corrected and normalised experimental data.

## 2. Materials and Methods

### 2.1. Sample Preparation and Standard Characterisation

Mg-Zn-Ca master alloys were prepared by arc-melting under vacuum (better than 3.0 × 10^−3^ Pa) mixing constituent elements of high purity (Mg 99.98 wt.%, Zn 99.9 wt.%, and Ca 99.5 wt.%). Ribbon specimens of the composition Mg_93−x_Zn_x_Ca_7_, where x = 3, 13, 23, 33, 43, 53, 63, and 73, were fabricated by ejecting molten master alloys under pressure of purified Ar through an orifice on the surface of a rotated cooper wheel. Ribbons were about 50 µm thick, 5 mm wide, and 100 mm long. The final chemical composition of the as-prepared ribbons was determined by energy-dispersive X-ray microanalysis, employing a scanning electron microscopy Jeol JSM 700F (JEOL Ltd., Akishima, Japan) with an accelerating voltage of 15 keV.

Mechanical properties such as elastic modulus and hardness were obtained using a nano-indentation tester, TTX-NHT S/N:01-03730 CSM Instruments (Lausanne, Switzerland), using a Berkovich pyramid diamond tip. In total, 20 indentations were performed and the final data were statistically evaluated.

The mass density of as-prepared materials was determined using a helium pycnometer, AccuPyc II 1340.

Thermal analysis measurements were performed using a Perkin-Elmer differential scanning calorimeter, DSC 8500 (PerkinElmer, Waltham, MA, USA), at a heating rate of 10 °C/min. The baseline was modelled using a polynomial function of the fifth order and then subtracted from the raw data.

### 2.2. Synchrotron-Based High-Energy X-ray Diffraction

High-energy X-ray diffraction (HEXRD) measurements were carried out at the high energy beamlines P21.1 [22] at Deutsches Elektronen-Synchrotron in Hamburg (Germany) and I12-JEEP [23] at Diamond Light Source in Didcot (United Kingdom). All diffraction measurements were performed in transmission geometry using a monochromatic X-ray beam of energy above 100 keV. Diffracted signals were detected by flat-panel detectors. The precise energy calibration was realised by collecting calibration data from a fine CeO_2_ standard at several standard-to-detector distances [24]. Once the X-ray beam energy was established, the detector was positioned at a desired position. Then, the CeO_2_ standard was measured again to calibrate absolutely the sample-to-detector distance, the orthogonality of a detector with respect to an incoming X-ray beam, and the position of a beam centre on the detector. All of those calibration procedures together with data integration along the radius of diffraction circles into *Q*-space (*Q* stands for the scattering vector) were performed employing the DAWN software [25]. Preferring *Q*-space over *2θ*-space (*2θ* is the scattering angle) reflects the definition of the magnitude of the scattering vector, *Q*, as *Q* = (4*π*/*λ*)sin(*θ*), which enables a direct comparison of diffraction patters collected using various wavelengths of a X-ray beam, *λ*. Raw data were corrected for background (air and container) contribution, self-absorption, fluorescence and Compton scattering, and normalised to electron [26].

For in situ high-temperature HEXRD measurements, sample heating from room temperature up to 550 °C at a heating rate of 10 °C/min was carried out using a commercial Linkam DSC600 furnace. The acquisition time of a diffraction image was 30 s. The temperature calibration was performed by measuring the Au powder sample and applying a state of equation derived by work presented elsewhere [27].

### 2.3. Neutron Diffraction

The neutron diffraction measurements were carried out at the 7C2 liquid and amorphous diffractometer of Laboratoire Léon Brillouin (LLB) in Saclay-Paris, France [28]. Measurements of V and Ni standard powder samples were firstly carried out to determine detector efficiency, wavelength, and detector position. The wavelength of incident neutrons was 0.724 Å. Specimens were loaded into vanadium sample holders of 6 mm in diameter. The correction of raw data for background scattering, multiple scattering, and detector efficiency followed established procedures [29].

### 2.4. X-ray and Neutron Structure Factor Calculation

Normalised elastically scattered X-ray and neutron intensities, *I^X^*(*Q*) and *I^N^*(*Q*), were used to calculate the total X-ray and neutron structure factors, *S^X^*(*Q*) and *S^N^*(*Q*), applying the Faber–Ziman formalism [30] as follows

for X-rays:


(1)
$$S^{X}\left(Q\right)=1+\left[I^{X}\left(Q\right)-\sum_{i}c_{i}f_{i}^{2}\left(Q\right)\right]/{\left[\sum_{i}c_{i}f_{i}\left(Q\right)\right]}^{2}\;$$


for neutrons:

(2)$$S^{N}\left(Q\right)=1+\left[I^{N}\left(Q\right)-\sum_{i}c_{i}b_{i}^{2}\right]/{\left[\sum_{i}c_{i}b_{i}\right]}^{2}\;$$
where *c_i_*, *f_i_*(*Q*), and *b_i_* are the atomic concentration, X-ray atomic scattering factor, and neutron coherent scattering length of the atomic species of type *i*, respectively. Then, the total X-ray and neutron reduced pair distribution functions, *D^X^*(*r*) and *D^N^*(*r*), were calculated as a Fourier sine transformation of corresponding structure factors as follows:(3)$$D^{M}\left(r\right)=\frac{2}{\pi }\int_{Q_{\min}}^{Q_{\max}}Q\left[S^{M}\left(Q\right)-1\right]\sin\left(rQ\right)dQ$$
where *Q*_min_ and *Q*_max_ are minimum and maximum values of the scattering variable *Q* in the analysis and *M* = *X* or *N*.

In the case of multicomponent systems following the Faber–Ziman formalism, the total structure factor *S^M^*(*Q*) can be expressed as a weighted combination of partial structure factors $S_{ij}^{M}\left(Q\right)$ defined for atom pairs coming from atoms of type *i* around atoms of type *j* as follows:(4)$$S^{X,N}\left(Q\right)=\sum_{i}\sum_{j}w_{ij}^{X,N}S_{ij}^{X,N}\left(Q\right)$$
with weights $w_{ij}^{X}$ and $w_{ij}^{N}$:(5)$$w_{ij}^{X}=\frac{c_{i}c_{j}f_{i}\left(Q\right)f_{j}\left(Q\right)}{{\left[\sum_{i}c_{i}f_{i}\left(Q\right)\right]}^{2}}\;and\;w_{ij}^{N}=\frac{c_{i}c_{j}b_{i}b_{j}}{{\left[\sum_{i}c_{i}b_{i}\right]}^{2}}$$

Then, the partial reduced pair distribution function can also be calculated as a sine Fourier sine transformation of the corresponding partial structure factor:(6)$$D_{ij}^{M}\left(r\right)=\frac{2}{\pi }\int_{Q_{\min}}^{Q_{\max}}Q\left[S_{ij}^{M}\left(Q\right)-1\right]\sin\left(rQ\right)dQ$$

*D^N^*(*r*) can be expressed as a combination of $D_{ij}^{N}\left(r\right)$:(7)$$D^{N}\left(Q\right)=\sum_{i}\sum_{j}w_{ij}^{N}D_{ij}^{N}\left(Q\right)$$

We note that, for *D^X^(r)*, Equation (7) is not strictly fulfilled because of the *Q*-dependence of *w*_ij_ weights. Nevertheless, the approximate equality can be used in a way similar to the case of *D^N^*(*r*).

## 3. Results

### 3.1. The As-Prepared State of the Alloys—Mechanical Properties

To start the characterisation of the as-prepared Mg-Zn-Ca specimens, their chemical compositions were verified by EDX. Small deviations between aimed and obtained chemical compositions were detected, especially for the concentration of Ca, which varies between 6 and 7 at.%. The mass density gradually increases from 1.72 g·cm^−3^ to 4.4 g·cm^−3^ as the compositions varies from Mg_90_Zn_3_Ca_7_ to Mg_30_Zn_63_Ca_7_. Mechanical properties such as elastic modulus and hardness were measured. It is confirmed that specimens prepared in an amorphous state have an elastic modulus in the range from 49 to 59 GPa and hardness between 3.6 and 5.0 GPa. All of the above-mentioned physical quantities are listed together for amorphous and crystalline Mg_93−x_Zn_x_Ca_7_ specimens in Table 1.

### 3.2. The As-Prepared State of the Alloys—Structural Characterisation

The X-ray scattering signals, *I^X^*(*Q*), collected from a series of as-prepared Mg_93−x_Zn_x_Ca_7_ specimens, are shown in Figure 1. Those HEXRD data clearly demonstrate a fully amorphous character of Mg_50_Zn_43_Ca_7_, Mg_60_Zn_33_Ca_7_, and Mg_70_Zn_23_Ca_7_ by the absence of detectable sharp Bragg peaks and the presence of a broad first diffuse peak. While only a tiny trace of a crystalline Mg phase is detected in an amorphous matrix of Mg_80_Zn_14_Ca_6_, its presence is significant in Mg_90_Zn_3_Ca_7_. The appearance of strong Bragg peaks in scattering signals of Mg_40_Zn_53_Ca_7_, Mg_30_Zn_63_Ca_7_, and Mg_20_Zn_73_Ca_7_ also excludes the glassy character of these samples. Those peaks are mainly assigned to a hexagonal Ca_4_Mg_13_Zn_29_ phase (184415-ICSD) marked as an IM3-type structure in [31]. In the case of Mg_20_Zn_73_Ca_7_, a hexagonal CaZn_11_ (184413-ICSD) phase is also identified. The experimental observation of the glass forming ability of the Mg_93−x_Zn_x_Ca_7_ alloys is compared to a prediction for an Mg-Zn-Ca system based on a machine learning approach [32]. A colour map shown in Figure 1 displays the largest probability of particular Mg-Zn-Ca ternary compositions to form a metallic glass in the red. Considering a composition range of studied specimens along a line with 7 at.% Ca, the highest predicted probability for a formation of MG is in the range from *x* = 20 to *x* = 50 for Mg_93−x_Zn_x_Ca_7_. This prediction is fully confirmed experimentally by X-ray diffraction data.

In order to extract information about a local atomic structure of identified amorphous Mg-Zn-Ca samples, structure factors and corresponding reduced pair distribution functions were calculated for Mg_50_Zn_43_Ca_7_, Mg_60_Zn_33_Ca_7_, and Mg_70_Zn_23_Ca_7_ using X-ray and neutron diffraction data, as shown in Figure 2. The structure factors are dominated by one diffuse peak referred to as a first diffraction peak (FDP) followed by a series of wigglers vanishing beyond 15 Å^−1^. However, in the case of Mg_70_Zn_23_Ca_7_, some sharper features in *S^N^*(*Q*) can be seen, indicating that the sample is not completely amorphous. The position of the FDP shifts to lower *Q*-values from 2.61 Å^−1^ to 2.53 Å^−1^ for *S^N^*(*Q*) and from 2.71 Å^−1^ to 2.52 Å^−1^ for *S^X^*(*Q*) as the atomic concentration of magnesium is increased.

The *D*(*r*) signal of each sample is quickly suppressed at higher *r* values, and only very weak oscillations are present beyond 10 Å, indicating the lack of long-range order. Furthermore, systematic changes in *D*(*r*)s with composition are detected. As can be seen in Figure 2c,d, positions of all of the oscillations including the first maximum shift to larger *r* values with the increase in Mg content. *D*(*r*) in the range of distances from 2 Å to 4 Å indicates the presence of only one maximum, but with a clear shape asymmetry, mainly detectable by X-ray data. The first maximum of *D*(*r*) is a subject of a particular interest as it contains direct structural information about the first coordination shell of an alloy. A shift of the maximum position to larger *r* values when magnesium substitutes zinc can be viewed as an expansion of the average interatomic distance of the first shell due to the replacement of a smaller Zn atom (1.332 Å) by larger Mg one (1.6 Å). On the other hand, the alteration of the total coordination number proportional to the atomic density and area under the first maximum is not significantly influenced when Mg atoms replace Zn ones, as listed in Table 2.

The interpretation of observed changes in *D*(*r*) is more challenging than one would anticipate. In a ternary system, six partial atomic pairs contribute simultaneously to the total *D*(*r*). The strength of individual contributions depends on the chemical composition of the alloy and an ability of the corresponding atoms to scatter X-rays and/or neutrons. In Figure 3a,b, the calculated X-ray and neutron weights of each partial reduced pair distribution function (see Equation (5)) of the investigated Mg-Zn-Ca alloys are displayed together with the part of total *D^X^*(*r*) and *D^N^*(*r*) functions corresponding to the first coordination shell. The position of atomic pairs was estimated on the basis of their metallic radii (*r*_Mg_ = 1.6 Å, *r*_Zn_ = 1.332 Å, *r*_Ca_ = 1.973 Å). The weights were calculated using the following values of X-ray atomic scattering factor (in electron units) and neutron coherent scattering length: *f_Mg_*(0) = 12 e, *f_Zn_*(0) = 30 e, *f_Ca_*(0) = 20 e, *b_Mg_* = 5.375 fm, *b_Zn_* = 5.68 fm, and *b_Ca_* = 4.7 fm. The atomic pairs contributing the most to both X-ray and neutron data are MgMg, MgZn, and ZnZn. Individual contributions of MgCa and ZnCa pairs to the total *D*(*r*) are weak and the effect of CaCa pairs on *D*(*r*) would be barely detectable either by X-rays or neutrons. The weight of MgMg pairs significantly rises as the amount of magnesium is increased in the sample, while the opposite behaviour is seen for ZnZn pairs. For clarity reasons, the X-ray and neutron weights of each of the studied Mg-Zn-Ca alloys are also listed in Table 3.

Figure 3c shows the difference between the reduced pair distribution function of Mg_50_Zn_43_Ca_7_ and Mg_70_Zn_23_Ca_7_ for both types of radiations. The subtraction of the data for the sample with a higher Mg content from the data for the sample with a lower Mg content should result in positive peaks due to zinc correlations and negative peaks due to magnesium correlations. The positions of the maxima and minima in this difference curve could serve as a first guess for the estimation of the interatomic distances between different constituents. The detected maximum at 2.67 Å and 2.71 Å for X-ray and neutron data, respectively, is associated with ZnZn pairs, while the minimum at 3.26 Å should be assigned to MgMg pairs. Considering theoretical metallic radii, the minima at 3.47 Å could be attributed to MgCa correlations.

In the following step, attempts are made to deconvolute the first peaks of *D^N^*(*r*) and *D^X^*(*r*) into three Gaussian functions. A single Gaussian function is defined as $G\left(x\right)=A\exp\left[-ln2{\left(\left(x-p\right)/\left(w/2\right)\right)}^{2}\right]$, where *A*, *p*, and *w* stand for maximum/peak parameters: amplitude, peak position, and full width at the half maximum. Individual Gaussians are interpreted as representations of ZnZn, MgZn, and MgMg pair contributions. As discussed above, other pairs are neglected owing to their weak contribution to the measured diffraction data. In addition, the datasets of *D^N^*(*r*) and *D^X^*(*r*) are fitted simultaneously using the same parameters for peak positions and broadenings to decrease their uncertainties. While fitting in the case of Mg_50_Zn_23_Ca_7_ and Mg_60_Zn_33_Ca_7_ was relatively straightforward, Mg_70_Zn_23_Ca_7_ required further restrictions. To obtain a stable fit, peak positions for the first and third Gaussians were fixed at 2.71 Å and 3.27 Å, respectively. These restraints are proposed in accordance with the analysis based on the difference curves presented in Figure 3c. The result of the whole deconvolution procedure is displayed in Figure 4 and the final fitted parameter values are listed in Table 4. It is seen that the final fitting curves match the first peaks of both *D^N^*(*r*) and *D^X^*(*r*) nicely for all three alloys. Generally, it can be claimed that the obtained fits seem to be reasonable with physical expectations. The contribution of the ZnZn Gaussian is more pronounced for *D^X^*(*r*) than for *D^N^*(*r*), reflecting that the weight of the ZnZn partial structure factor is higher in the X-ray dataset (Table 3). Finally, an opposite tendency is observed for the MgMg Gaussian when *D^N^*(*r*) has a more pronounced contribution than *D^X^*(*r*), again mirroring the higher sensitivity of neutron data to Mg-related correlations.

The MgMg nearest neighbour distance of 3.27 Å obtained here could be viewed as larger compared with values proposed by other studies of similar systems, suggesting 3.12 Å [33], 3.00–3.05 Å [34], 3.025 Å [35], or 3.10 Å [36], mainly based on molecular dynamics (MD) simulations. One may advocate that a shift of MgMg Gaussian to larger *r* values is just a consequence of neglecting contributions of ZnCa, MgCa, and CaCa pairs trying to compensate for their absence in the model. On the other hand, it was demonstrated for amorphous Mg_72_Zn_28_ using X-ray and neutron diffraction data that the MgMg pair distance is 3.2 Å, a value seemingly larger than that proposed by modelling. Values of ~2.7 Å and ~2.95 Å for ZnZn pairs and MgZn pairs, respectively, are within the range of values proposed by theoretical works [33,35,36].

### 3.3. The Thermal Evolution of the Glassy Alloys

The thermal stability of Mg_50_Zn_43_Ca_7_, Mg_60_Zn_33_Ca_7_, and Mg_70_Zn_23_Ca_7_ was inspected by DSC measurements in the range of ambient temperature and 550 °C. The DSC curves displayed in Figure 5 show the presence of multiple exothermic events (their onsets are marked by T_x1_, T_x2_, T_x3_, and so on), suggesting a complex devitrification process and one endothermic event T_m_ associated with melting. In the case of Mg_70_Zn_23_Ca_7_ and Mg_60_Zn_33_Ca_7_, the first exothermic event is clearly detected as a small exothermic peak with the onset T_x1_ at ~87 °C followed by more pronounced second exothermic peaks T_x2_ at 128 °C and 172 °C, respectively. the Mg_50_Zn_43_Ca_7_ DSC curve shows a shallow minimum appearing around T_X1_ ~130 °C followed by a sharp exothermic peak starting at T_X2_ ~205 °C, clearly indicating a crystallisation event. It can be concluded that a gradual substitution of Mg by Zn within the Mg_93−x_Zn_x_Ca_7_ system inhibits a first crystallisation event and leads to a better thermal stability at low temperatures. The melting temperature slightly decreases from 353 °C to 326 °C as a function of the increase in Zn.

In addition to DSC, in situ high temperature HEXRD measurements were performed for Mg_60_Zn_33_Ca_7_ and Mg_50_Zn_43_Ca_7_. Owing to technical issues during the beamtime, in situ data were not collected for Mg_70_Zn_23_Ca_7_. Diffraction patterns obtained uninterruptedly for the duration of the heating process from room temperature to 520 °C with a resolution of 5 °C per pattern are shown in Appendix A as structure factor and reduced pair distribution function curves. Alike DSC, X-ray diffraction data confirm a complex temperature evolution of both Mg_60_Zn_33_Ca_7_ and Mg_50_Zn_43_Ca_7_ glasses. Firstly, structure factors possess a smooth character, demonstrating a glassy state of the samples. In the temperature range between 90 °C and 170 °C, only tiny changes in the region of the FDP of S(Q) are detectable in the form of a continuous formation of FDP splitting; see Figure 6. Those changes propose the formation of a minor nanocrystalline phase, but its phase identification was not possible at this stage by available XRD data. It is worthy to note that, observing corresponding *D*(*r*) curves, no changes are observed up to 4 Å (the first coordination shell). It can be proposed that the nearest atomic arrangement of a formed minor nanocrystalline phase could be very close to a local atomic arrangement of the amorphous matrix. The modest evolution of HEXRD patterns collected at low temperatures (between ~90 and 170 °C) is in a good accordance with the DSC data presented above, indicating a faint exothermic event in the same temperature interval.

An undoubted formation of Bragg peaks is detected above ~164 °C and 201 °C for Mg_60_Zn_33_Ca_7_ and Mg_50_Zn_43_Ca_7_, respectively. It is clearly seen that some Bragg peaks appearing at an earlier stage later vanish and/or transform to new ones. At higher temperatures (above 350 °C), all Bragg peaks disappear, and diffraction patterns again have a diffuse profile, reflecting the liquid state of the samples. Crystallisation events observed by HEXRD are close to those detected by calorimetry measurements. They are compared and listed together in Table 5.

The crystallisation processes of a similar system, Mg_72−x_Zn_24+x_Ca_4_ for *x* = 0, 2, 4, 8, 10, 12, and 14, have previously been investigated in detail using ex situ XRD by Zhang et al. [31]. They identified the formation of a cubic Mg_51_Zn_20_ phase at the early stage followed by the precipitation of an hcp-Mg phase and intermetallic compound IM1 (Ca_3_Mg_x_Zn_15−x_ for 4.6 ≤ *x* ≤ 12 with Sc_3_Ni_11_Si_3_ prototype [37]) from the remaining amorphous matrix. At higher temperatures, the Mg_51_Zn_20_ phase disappeared and two other intermetallic compounds IM3 and IM4 were formed. Finally, before melting, the devitrification process ended with phases of hcp-Mg, IM1, and IM3 [31]. Phase analysis of our specimens, Mg_60_Zn_33_Ca_7_ and Mg_50_Zn_43_Ca_7_, was hampered by low angular resolution of the HEXRD setup, optimised for the study of amorphous systems. Selected diffraction patterns of Mg_60_Zn_33_Ca_7_ and Mg_50_Zn_43_Ca_7_ at different stages of their devitrification process are displayed in Figure 7. Nevertheless, we are able to confirm the precipitation of IM1-type and Mg phases during the second crystallization, and later followed by the formation of an IM3-type phase similar to Ca_4_Mg_13_Zn_29_ (184415-ICSD). The appearance and disappearance of an additional phase during the third crystallisation event for Mg_60_Zn_33_Ca_7_ was also observed. It is reasonable to identify it with the IM4 compound reported by Zhang et al. [31]. In contrast to the previous work, the crystallisation of Mg_51_Zn_20_ at the beginning of the devitrification process was not distinguished. Finally, comparing a diffraction pattern of Mg_60_Zn_33_Ca_7_ and Mg_50_Zn_43_Ca_7_ after all crystallisation events and before melting, it seems that the higher Zn concentration resulted in a more pronounced presence of the IM1 phase (see Figure 7).

## 4. Conclusions

It was demonstrated that amorphous ribbon alloys of the Mg_93−x_Zn_x_Ca_7_ system can be fabricated in the range of 20 < x < 50 at.%. The structure of as-prepared glassy alloys Mg_70_Zn_23_Ca_7_, Mg_60_Zn_33_Ca_7_, and Mg_50_Zn_43_Ca_7_ was investigated by means of atomic pair distribution analysis employing high-energy X-ray diffraction and neutron diffraction data. While the first maximum of reduced pair distribution functions gradually modifies its shape and shifts to larger *r* values with the decrease in Zn concentration, the total coordination number remains close to 12, without a significant change. Simultaneous fitting of the first peaks of *D^X^*(*r*) and *D^N^*(*r*) functions by three Gauss functions enables to approximate the contributions of MgMg, MgZn, and ZnZn pairs. Their nearest interatomic distances were estimated to be ~3.27 Å, ~2.95 Å, and ~2.7 Å. In order to determine the contributions of the remaining pairs (MgCa, ZnCa, and CaCa), additional experimental data would be required, involving techniques such as Ca *K*-edge extended X-ray absorption spectroscopy or neutron diffraction with isotopic substitutions, bringing their own challenges in data collection and sample preparation. The basic mechanical characterisation of the as-prepared amorphous specimens revealed a weak systematic influence of the chemical composition on the elastic modulus and hardness. The devitrification process of the samples was investigated by DSC and in situ HEXRD measurements. For all three glassy alloys (Mg_70_Zn_23_Ca_7_, Mg_60_Zn_33_Ca_7_ and Mg_50_Zn_43_Ca_7_), a complex crystallisation path was observed, exhibiting at least four crystallisation events. It was observed that the first crystallisation was inhibited with the increase in Zn content. At earlier stages of devitrification, the formation of hcp-Mg and intermetallic IM1 phases was recognised, followed by the precipitation of the intermetallic IM3 phase. For future work, Mg_70_Zn_23_Ca_7_, Mg_60_Zn_33_Ca_7_, and Mg_50_Zn_43_Ca_7_ could be used as precursors for microalloying with the aim to improve their mechanical properties (e.g., plasticity).

## Figures and Tables

**Figure 1 materials-16-02313-f001:**
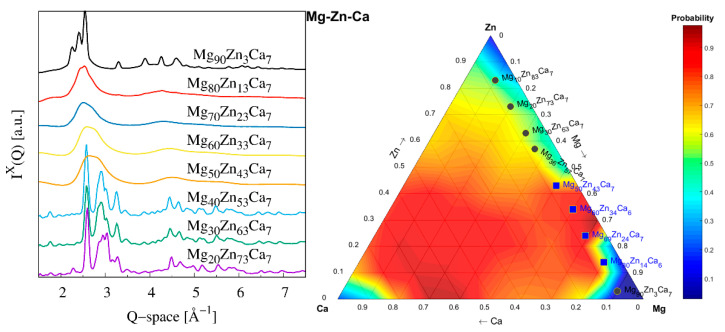
(**left**) The X-ray diffraction patterns of the as-prepared Mg_93−x_Zn_x_Ca_7_ specimens indicated by legend (intensity curves are vertically offset for clarity reasons) and (**right**) the Mg-Zn-Ca ternary colourmap showing a prediction of the glass forming probability calculated by a machine learning algorithm. The alloys prepared in this work identified as crystalline are marked by circles and those identified as amorphous are labelled by squares.

**Figure 2 materials-16-02313-f002:**
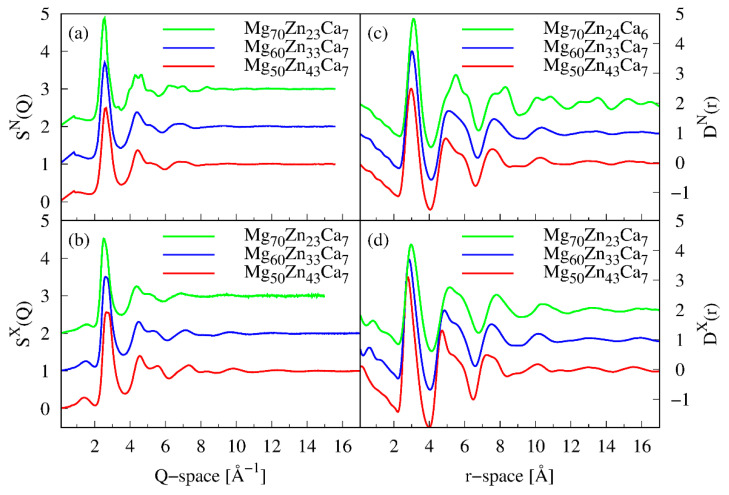
(**a**) Neutron and (**b**) X-ray structure factors and corresponding (**c**) neutron and (**d**) X-ray reduced pair distribution functions for Mg_70_Zn_23_Ca_7_, Mg_60_Zn_33_Ca_7_, and Mg_50_Zn4_3_Ca_7_ (from top to bottom). Individual curves are vertically offset for clarity reasons.

**Figure 3 materials-16-02313-f003:**
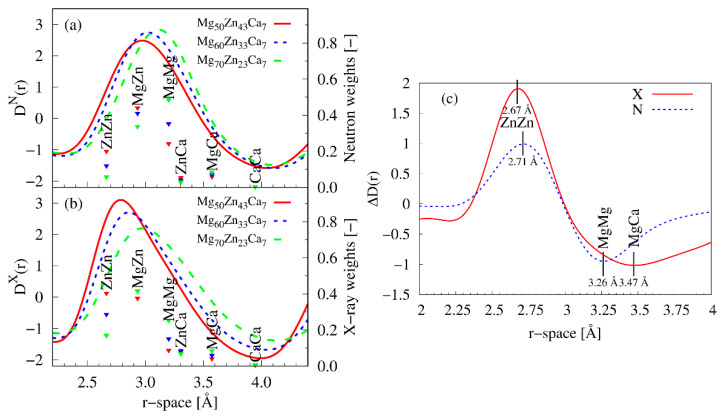
The first maxima of (**a**) neutron and (**b**) X-ray reduced total pair distribution functions corresponding to the first coordination shells of Mg_50_Zn4_3_Ca_7_, Mg_60_Zn_33_Ca_7_, and Mg_70_Zn_23_Ca_7_ together with neutron and X-ray weights (at *Q* = 0 Å) of the partial reduced pair distribution functions. (**c**) The difference curve, Δ*D*(*r*), between the total reduced pair distribution functions of Mg_60_Zn_33_Ca_7_ and Mg_50_Zn_43_Ca_7_ for neutrons (N) and X-rays (X) in the range of distances of the first coordination shell.

**Figure 4 materials-16-02313-f004:**
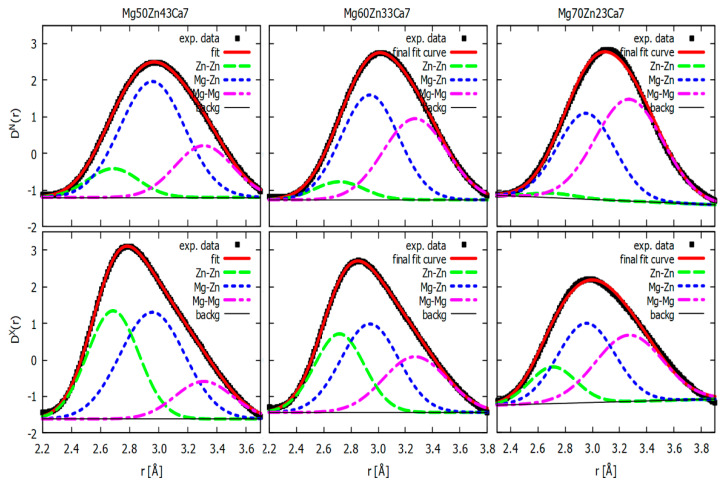
Decomposition of the first *D^X^*(*r*) and *D^N^*(*r*) into three Gaussians representing ZnZn, MgZn, and MgMg pairs for Mg_50_Zn_43_Ca_7_, Mg_60_Zn_33_Ca_7_, and Mg_70_Zn_23_Ca_7_.

**Figure 5 materials-16-02313-f005:**
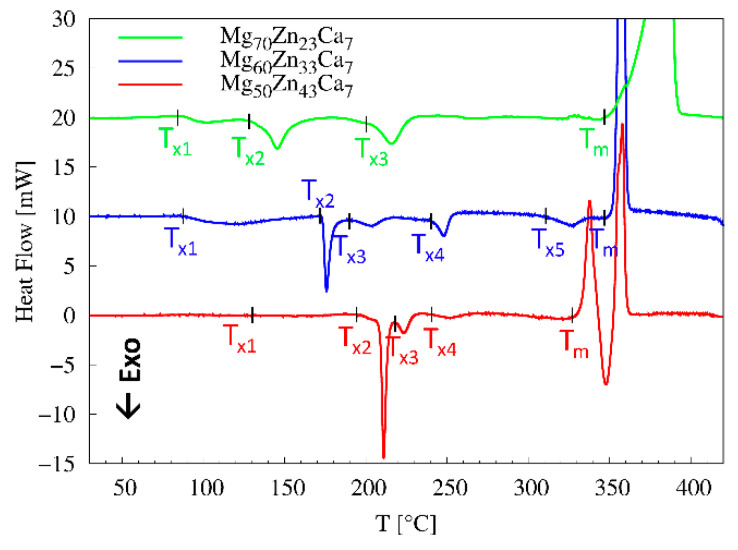
The DSC curves for Mg_70_Zn_23_Ca_7_, Mg_60_Zn_33_Ca_7_, and Mg_50_Zn_43_Ca_7_ (from top to bottom). Individual curves are vertically offset for clarity reasons.

**Figure 6 materials-16-02313-f006:**
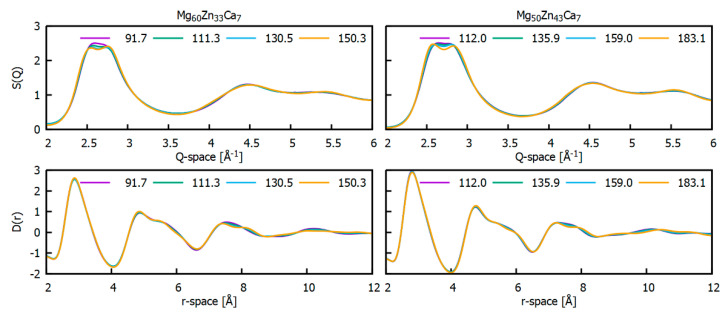
Selected structure factor and corresponding reduced pair distribution functions, *S*(*Q*) and *D*(*r*), collected at different temperatures during a first crystallisation event.

**Figure 7 materials-16-02313-f007:**
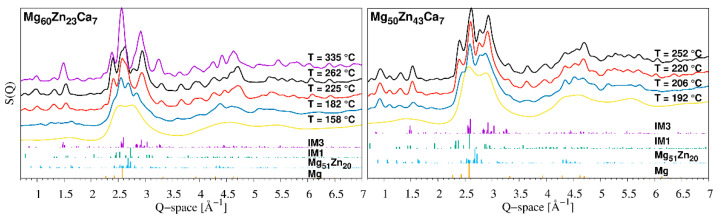
The selected *S*(*Q*) curves for Mg_60_Zn_33_Ca_7_ (left and Mg_50_Zn_43_Ca_7_ (right) obtained at different temperatures (indicated by legend) together with relative intensity plots of proposed crystalline phases IM3, IM1, Mg_51_Zn_20_, and hcp-Mg from top to bottom (indicated by legend). Individual *S*(*Q*) curves are vertically offset for clarity reasons.

**Table 1 materials-16-02313-t001:** Chemical composition, phase, mass density, and corresponding calculated atomic mass density, hardness, and elastic modulus of the as-prepared Mg_93−x_Zn_x_Ca_7_ alloys.

Sample Name	EDX [at.%]	Phase	Mass Density [g/cm^3^]	Atomic Mass Density [atoms/Å^3^]	H [GPa]	E [GPa]
Mg_90_Zn_3_Ca_7_	Mg_91_Zn_3_Ca_6_	crystalline	1.72 ± 0.01	0.0391	3.1 ± 0.3	24.8 ± 2.5
Mg_80_Zn_13_Ca_7_	Mg_80_Zn_14_Ca_6_	crystalline	2.13 ± 0.01	0.0417	3.1 ± 0.3	42.0 ± 2.3
Mg_70_Zn_23_Ca_7_	Mg_69_Zn_24_Ca_7_	amorphous	2.27 ± 0.02	0.0385	3.6 ± 0.4	49.7 ± 3.5
Mg_60_Zn_33_Ca_7_	Mg_60_Zn_34_Ca_6_	amorphous	2.69 ± 0.01	0.0414	4.2 ± 0.8	52.9 ± 6.0
Mg_50_Zn_43_Ca_7_	Mg_50_Zn_43_Ca_7_	amorphous	2.93 ± 0.01	0.0410	5.0 ± 0.1	58.2 ± 0.9
Mg_40_Zn_53_Ca_7_	Mg_36_Zn_57_Ca_7_	crystalline	4.21 ± 0.02	0.0519	-	-
Mg_30_Zn_63_Ca_7_	Mg_30_Zn_63_Ca_7_	crystalline	4.40 ± 0.02	0.0517	6.9 ± 0.5	74.2 ± 4.6

**Table 2 materials-16-02313-t002:** Total coordination numbers, *N^X^* and *N^N^*, calculated in the range of 2.3 Å to 4.0 Å from *D^X^*(*r*) and *D^N^*(*r*) functions, respectively.

Sample	*N^X^*	*N^N^*
Mg_70_Zn_24_Ca_7_	11.2	11.9
Mg_60_Zn_33_Ca7	11.4	12.3
Mg_50_Zn_43_Ca_7_	11.5	12.0

**Table 3 materials-16-02313-t003:** X-ray (at *Q* = 0 Å) and neutron weight coefficients, *w^X^* and *w^N^*, of each partial reduced pair distribution function of Mg_50_Zn_23_Ca_7_, Mg_60_Zn_33_Ca_7_, and Mg_70_Zn_23_Ca_7_.

	MgMg		MgZn		MgCa		ZnZn		ZnCa		CaCa	
	*w^X^*	*w^N^*	*w^X^*	*w^N^*	*w^X^*	*w^N^*	*w^X^*	*w^N^*	*w^X^*	*w^N^*	*w^X^*	*w^N^*
Mg_50_Zn_23_Ca_7_	0.087	0.242	0.376	0.441	0.041	0.059	0.404	0.200	0.088	0.054	0.005	0.004
Mg_60_Zn_33_Ca_7_	0.152	0.353	0.417	0.410	0.059	0.072	0.286	0.119	0.081	0.042	0.006	0.004
Mg_70_Zn_23_Ca_7_	0.253	0.486	0.416	0.337	0.084	0.085	0.171	0.059	0.069	0.030	0.007	0.004

**Table 4 materials-16-02313-t004:** Fitted Gaussian parameters representing ZnZn, MgZn, and MgMg pairs for Mg_50_Zn_43_Ca_7_, Mg_60_Zn_33_Ca_7_, and Mg_70_Zn_23_Ca_7_. Errors of individual parameters correspond to the fitting errors.

	“ZnZn” Gaussian				“MgZn” Gaussian				“MgMg” Gaussian			
	*p* _1_	*w* _1_	*A* _1_ * ^X^ *	*A* _1_ * ^N^ *	*p_2_*	*w* _2_	*A* _2_ * ^X^ *	*A* _2_ * ^N^ *	*p* _3_	*w* _3_	*A* _3_ * ^X^ *	*A* _3_ * ^N^ *
Mg_50_Zn_43_Ca_7_	2.687(2)	0.414(2)	3.0(1)	0.8(1)	2.96(1)	0.52(2)	2.9(1)	3.2(1)	3.31(2)	0.47(2)	1.0(2)	1.4(2)
Mg_60_Zn_33_Ca_7_	2.717(5)	0.426(5)	2.1(2)	0.5(2)	2.94(2)	0.50(2)	2.4(3)	2.9(4)	3.27(4)	0.55(3)	1.5(3)	2.2(3)
Mg_70_Zn_23_Ca_7_	2.71 *	0.40(1)	1.0(2)	0.1(2)	2.96(1)	0.49(3)	2.2(1)	2.4(1)	3.27 *	0.58(1)	1.8(1)	2.8(1)

* fixed during the fitting.

**Table 5 materials-16-02313-t005:** Crystallisation temperatures determined by DSC and in situ HEXRD measurements at a heating rate of 10 °C per minute.

	T_X1_ [°C]		T_X2_ [°C]		T_X3_ [°C]		T_X4_ [°C]		T_X5_ [°C]		T_m_ [°C]	
Alloy	DSC	XRD	DSC	XRD	DSC	XRD	DSC	XRD	DSC	XRD	DSC	XRD
Mg_70_Zn_23_Ca_7_	87	-	128	-	187	-		-		-	347	-
Mg_60_Zn_33_Ca_7_	89	~91	172	164	194	187	241	235	312	296	350	340
Mg_50_Zn_43_Ca_7_	~130	~122	205	201	219	211	240	230	-	-	327	315

## Data Availability

Data will be made available upon request.

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
