# Peer review of "Structure and Physical Properties of Mg_93−x_Zn_x_Ca_7_ Metallic Glasses"

_materials, 2023, doi:10.3390/ma16062313_

Round 1

Reviewer 1 Report

The paper is well done by the authors. However, some suggestions are illustrated as follows before the acceptance for the publication:

(1) In the introduction part, the state-of-the-art of Mg-Zn-Ca metallic glasses should be added. Moreover, based on the summary on the research progress of Mg-Zn-Ca MGs, the authors can give their novelty of this paper. Currently the readers are not easy in understanding the novelty of this paper.

(2) For Fig.6 and Fig.7, certainly including Table 5, why did the authors not give the corresponding data and analysis on Mg70Zn23Ca7 MGs? If discussing the effect of Zn contents, it is suggested that Mg70Zn23Ca7 should be given the corresponding description and discussion.

(3) Please check the English carefully.

Author Response

Dear Reviewer, please find our responses in the attachment.

Reviewer 2 Report

The manuscript is devoted to the study of the structure of amorphous alloys and its dependence on composition. The article contains a number of new interesting results and may be published after some corrections. 

- Line 140: The alloy compositions are clearly incorrect (Mg50Zn43Ca7, Mg60Zn43Ca7)

-  The baseline was modelled using a polynomial function of the 5th order and 89 then subtracted from the raw data.  Usually an envelope is used for this. Why was the polynomial chosen in this case?

- The thicknesses of the obtained samples are not indicated.

- The authors state that the Mg50Zn43Ca7, Mg60Zn33Ca7 and Mg70Zn23Ca7 samples are amorphous. However, in fig. 1, the diffuse peak of the Mg70Zn23Ca7 alloy is obviously not symmetrical, and the diffuse reflections of the Mg60Zn33Ca7, Mg50Zn43Ca7 alloys have a very wide top of peak, which suggests a possible superposition of two diffuse maxima or diffuse and diffraction low intensity.  This should be explained.

- The authors have carried out a serious analysis of pair correlation functions. This causes great respect, this approach is not common. However, this part requires additional explanation, since the approach used is applied to binary alloys. In principle, a number of pairs should be considered: the environment of A atoms around an A atom (AA), B atoms around an A atom  (AB), C atoms around an  A atom(AC), A atoms around an B atom  (BA), etc. It should be noted that the pairs AB and BA (and other related pairs) are not equivalent. The application of the approach described in the article for the ternary alloy should be explained.

The article makes a good impression and can be published after correction

Reviewer 3 Report

The authors investigated the structure and physical properties of Mg-Zn-Ca amorphous alloy. The presented work is of acceptable quality, however, the following improvements can improve the manuscript to qualify for publishing in Materials.

1. the equations are inserted as images which have low resolution. It is preferable to insert them using the equation tool.

2. We suggest adding sub-titles to Section 2. Please order the results according to the methodologies.

3. Section 3.1: It is preferable to interpret how these results would affect the function of the Mg-Zn-Ca alloy.

4. Section 3.2: How can you decide the suitability of the mechanical properties? You need to justify your claims.

5. You may include future work if possible

Round 2

Reviewer 2 Report

The authors have made the necessary corrections, the manuscript can be published.